# Natural Blues: Structure Meets Function in Anthocyanins

**DOI:** 10.3390/plants10040726

**Published:** 2021-04-08

**Authors:** Alan Houghton, Ingo Appelhagen, Cathie Martin

**Affiliations:** 1John Innes Centre, Department of Metabolic Biology and Biological Chemistry, Norwich Research Park, Norwich NR4 7UH, UK; Alan.Houghton@jic.ac.uk; 2Phyton Biotech, Alter Postweg 1, 22926 Ahrensburg, Germany; Ingo.Appelhagen@gmx.de

**Keywords:** blue, anthocyanin, food colourants, co-pigment, quinonoid base

## Abstract

Choices of blue food colourants are extremely limited, with only two options in the USA, synthetic blue no. 1 and no. 2, and a third available in Europe, patent blue V. The food industry is investing heavily in finding naturally derived replacements, with limited success to date. Here, we review the complex and multifold mechanisms whereby blue pigmentation by anthocyanins is achieved in nature. Our aim is to explain how structure determines the functionality of anthocyanin pigments, particularly their colour and their stability. Where possible, we describe the impact of progressive decorations on colour and stability, drawn from extensive but diverse physico-chemical studies. We also consider briefly how this understanding could be harnessed to develop blue food colourants on the basis of the understanding of how anthocyanins create blues in nature.

## 1. Introduction

Throughout history, our culture has been intrinsically linked to the colours of pigments and dyes accessible to us [1]. Colour is one of the primary visual indicators of food spoilage, so it comes as no surprise that we have evolved with innate psychological responses to abnormally coloured food; effects so strong that food not matching colour expectations can taste less appealing than an otherwise identical, “colour-correct” foods [2]. With recent trends towards use of natural pigments in food, manufacturers have invested heavily in developing replacements for synthetic colourants. Despite plenty of options for yellow to purple, natural blue colourants remain elusive [3].

Currently, only three synthetic blue pigments are approved in Europe as food additives. Brilliant blue FCF (derived from petrochemicals), Indigo carmine (synthetic derivative of indigo), and patent blue V. However, only one natural pigment is available—phycocyanin, derived from “Spirulina”, which is composed of a mixture of three species of cyanobacteria, namely, *Arthrospira platensis*, *Arthrospira fusiformis*, and *Arthrospira maxima*.

Natural blue pigments are rare in nature, and only one known animal, a species of butterfly, *Nessaea obrinus*, can synthesise them. Instead, many apparently blue organisms create multi-layered nanostructures to reflect shorter wavelength light selectively, giving them their iridescent blue hues [4]. Unlike pigments however, the structural colour will change in these cases, depending on the viewing or illumination angle.

Plants produce an array of natural colourants, which can be broadly separated into four major groups: chlorophyll (green), carotenoids (yellow-red), betalains (yellow and red), and flavonoids (yellow-blue). Plants utilise these pigments to perform essential functions such as photosynthesis, radical scavenging, and attracting pollinators. Many fruits and vegetables are coloured with red and orange pigments, which have the greatest contrast against green foliage. This increases the likelihood they will be seen and eaten, allowing their seeds to be dispersed widely. This seed dispersal strategy is so effective that three unique red pigments have evolved: lycopene (a carotenoid found in fruits such as tomatoes), betacyanins (betalains found in some of the *Caryophyllales*) [5], and anthocyanins (such as the major pigment in strawberries, pelargonidin 3-*O*-glucoside) [6].

Despite the wide variety of pigments synthesised by plants, few blossoms or fruits are blue. In fact, some estimates suggest only 10% of angiosperm species include blue varieties. A major difficulty in producing blue pigments arises, in part, due to how pigments are coloured in the first place. Electrons in the highest occupied molecular orbital of chromophores (HOMO) can absorb a discrete amount of energy (corresponding to a specific wavelength of light) which promotes an electron to the lowest unoccupied molecular orbital (LUMO) [7]. The light reflected is a mixture of all unabsorbed wavelengths, which creates the colours we see. Alternating π bonds increase the delocalisation of electrons, reducing the energy an electron requires to move from the ground state to the excited state [8]. As the HOMO-LUMO energy gap shrinks, longer wavelength (lower energy) light is absorbed, causing a bluing in colour. To produce true blues, the absorbed wavelength of light is relatively long at around 580–620 nm.

Compounds with the conjugated systems necessary to absorb light in this range are typically large and complex, requiring multiple biosynthetic steps and heavy energy investment in their biosynthesis. Compounding this problem, blue colouration provides only a relatively niche evolutionary advantage in attracting specific pollinators [9]. Combined, these are the principal reasons that blue pigments are relatively uncommon in nature. However, blue-flowered varieties have been heavily selected by breeders and horticulturalists, increasing the range of blue flowers available to enthusiastic gardeners.

Synthetic colourants have been used historically to colour or colour-correct within the food industry. The safety of these colourants has rightfully been questioned, starting with the Food and Drug Act of 1906 in the USA, and more recently promoting a shift towards natural sources of pigment. The replacement of all colours with natural alternatives has proved challenging, particularly for blue pigments, due to their increased sensitivity to light, temperature, and oxygen, as well as their low water solubility compared to synthetic dyes.

However, a diverse array of stunning natural blue shades are available, particularly in floral systems, based on pigmentation by anthocyanins. Typically, anthocyanins are unstable in solution, especially under conditions where blue colouration could potentially be observed. This is due to the rapid formation of colourless forms, and the eventual isomerisation into yellow forms, leading to browning of products. To stabilise the anthocyanin quinonoid base and generate blue colours, anthocyanins form complexes through intramolecular co-pigmentation, intermolecular co-pigmentation, and metal chelation. Many of these strategies have been adopted in nature by plants that produce blue flowers or fruit.

The mechanisms responsible for determining the hue and stability of colour of “blue anthocyanins” are complex and incompletely understood. The aim of this review is to summarise the physiochemical properties associated with blue pigmentation, illustrated by some key examples. We aim also to outline how this understanding might be used to formulate or engineer more effective natural blues to expand the pallet of natural food colorants.

## 2. General Features of Anthocyanins

Anthocyanins are a subclass of flavonoids; polyphenolic derivatives consisting of a fused benzopyrylium core (C- and A-rings) with an additional phenyl group attached at C2 (B-ring) formed by condensing a C6-C3 unit (*p*-coumaroyl-CoA) with three molecules of malonyl CoA (Table 1).

Twenty-seven naturally occurring anthocyanidins have been identified to date, however, six account for approximately 92% of all anthocyanins reported [10]. These six are pelargonidin, cyanidin, delphinidin, peonidin, petunidin, and malvidin (Table 1). The distributions of these six anthocyanidin classes in fruits and vegetables are cyanidin 50%, delphinidin 12%, pelargonidin 12%, peonidin 12%, petunidin 7%, and malvidin 7% [11].

Anthocyanidins are really only stable at very acid pH values (around pH 1); at physiological pH, they are quickly hydrated to form colourless forms (hemiketals). They are glycosylated at C3 in order to stabilise natural anthocyanidins, and these compounds are termed anthocyanins. Further decoration of the anthocyanin with carbohydrate, aromatic, and aliphatic moieties produces the wide array of anthocyanins seen in nature. It is these compounds which are responsible for almost all the orange/blue colours of floral systems.

Substitution of the B-ring with either hydroxyl or methoxyl groups influences the absorbance maxima of the anthocyanidin, and hence alters their colour from orange (pelargonidin) to bluish-purple (malvidin). Blue flowers commonly contain delphinidin derivatives, although there are a few exceptions where cyanidin-based anthocyanins confer blue [12].

The addition of hydroxyl groups on the B-ring tends to increase the maximum absorption of the anthocyanins in the visible range (λ_max_ in nm) under specified conditions. Thus, for pelargonidin 3-glucose, cyanidin 3-glucoside, and delphinidin 3-glucoside, the respective visible λ values are 510, 530, and 543 nm in 1% HCl in methanol [13].

It is thought that due to the electronegativity of the oxygen atom in the B-ring, substituent groups would pull electron density away from the aromatic centre. The additional substituent groups in the ortho and para positions of the B-ring have a lone pair of electrons present on the oxygen atom and are adjacent to the sp^2^ orbital of the B-ring. Electrons are donated through resonance with the C2-C1’ bond, which becomes more electron-rich in the presence of ortho and para hydroxyl groups on the B-ring [14]. This increased electron density increases the energy of both the ground state (HOMO) and excited states (LUMO). As these interactions occur through the non-bonding orbital (which is energetically closer to the HOMO), the ground state experiences a greater magnitude of effect, effectively shrinking the transition energy gap and promoting absorption of longer wave-length light and a bathochromic shift.

## 3. Anthocyanin Multistate Equilibrium

All features of anthocyanin pigmentation need to be explained in the context of the multistate equilibrium which exists for anthocyanins in solution. Anthocyanins function as weak diacids due to their chromophore containing a strongly electron-withdrawing pyrilium ring, giving the phenolic hydroxyl groups at C4’, C5, and C7 fairly acidic properties. Of these hydroxyl groups, the C7-OH is the most acidic (pKa1 ≈ 4), followed by the C4’ hydroxyl (pKa2 > 7) for simple anthocyanins [15]. As weak diacids, anthocyanins become deprotonated as pH increases which, in turn, increases electron polarisation and increases the absorption maxima towards higher wavelengths.

Anthocyanin colour therefore is the result of the ratio of the three coloured species in equilibrium at any specific pH. At very acidic pH values (around pH < 2), the formation of the red flavylium ion (AH^+^) is favoured (Figure 1). This species is fully protonated and has a delocalised positive charge across the chromophore. As the pH increases above pK_a1_, the first deprotonation occurs, converting flavylium ions into the neutral quinonoid base (A) and the colour changes from red to purple. At pH values above pK_a2_, the quinonoid base is deprotonated further, forming the anionic quinonoid base (A^−^) with a negative delocalised charge [16] (Figure 1). This deprotonation sequence is accompanied by two shifts in λ_max_, the first (AH^+^→ A) typically of 20–30 nm and the second (A→ A^−^) a further shift of 50–60 nm [15].

Due to the instability of A (and A^−^), for simple anthocyanins, the conversion of AH^+^ to A is generally less thermodynamically favourable than hydration at C2, giving the colourless hemiketal (B), and tautomerisation of B to give colourless isomeric chalcones (C_t_ and C_c_). As the most stable products above pH 3, species B and chalcones deplete the available pool of the coloured species, and the red or purple colours begin to disappear as the pH is increased.

A beautiful example of how vacuolar pH can impact flower colour has been described for Japanese morning glory, which presents reddish brown flower buds that change to a brilliant blue as the petals unfold. This colour change accompanies an increase in pH of the vacuoles from 6.6 in buds to an unusually high value of 7.7 in mature petals. The increase in vacuolar pH is conferred by a tonoplast-located Na^+^/H^+^ exchanger [17,18], responsible for increasing the vacuolar pH by exchanging H^+^ for Na^+^ in vacuoles of petal cells during flower opening.

## 4. Co-Pigmentation

The diversity of colours observed in floral systems cannot be rationalised by differences in chemical substituent groups and the pH equilibrium alone; without interactions that stabilise the chromophore at physiological pH within the vacuole (which is usually about pH 5.5), most flowers would be devoid of their attractive hues and would instead be colourless or pale yellow (as a result of hemiketal and chalcone formation, respectively; Figure 1). Anthocyanins are large, planar compounds which lend themselves to the formation of non-covalent interactions with molecules termed co-pigments (Figure 2). Generally, these interactions are favourable towards the stabilisation of the coloured species through blocking hydration of the flavylium cation to the colourless hemiketal by the addition of water at C2 [19]. These interactions occur with the highest affinity between the coloured forms of the anthocyanin species and the co-pigment, and less so with the hemiketal and chalcone forms due to decreased conjugation of the tricyclic core upon ring opening [15] (Figure 2). Observations from the anthocyanins found typically in red wines have demonstrated these co-pigmentation interactions occur most strongly at around pH 3.6, or when A is the dominant species [20].

### 4.1. Self-Assocation

Anthocyanins can also form self-association interactions with other anthocyanins. However, these complexes are typically less strongly associated than intermolecular co-pigmentation, requiring a relatively high concentration of anthocyanins before they are observed (around 1 mM) [21]. Self-association interactions are strongest between neutral species and are destabilised by charge repulsion between pairs of positively charged flavylium cations and between negatively charged anionic bases [15] (Figure 3).

This self-associative behaviour is dependent on the substitutions present on the B-ring. It has been shown that the self-association constant decreases in the series delphinidin 3-*O*-glucoside > pelargonidin 3-*O*-glucoside > cyanidin 3-*O*-glucoside. The addition of a hydroxyl group to the C3’ position introduces slight torsion into the B-ring: C-ring linkage, reducing the planarity and thus the proximity at which the cyanidin molecules can associate.

The opposite effect is observed with methylation where self-association increases from pelargonidin 3-*O*-glucoside < peonidin 3-*O*-glucoside < malvidin 3-*O*-glucoside. The steric hinderance of the methoxy groups on the B-ring may dictate the orientation of the complex, favouring associations with better alignment.

Decoration with sugars also influences the self-association stacking arrangement. Although not aromatic, the sugars are heteroatoms, with electron density polarised towards the oxygen. This property may induce further polarisation in the interacting molecule, strengthening the association, but the position and size of the sugar determines the overall alignment of the complex. Glucosylation at C3 and C5 of delphinidin increases the propensity for self-association, likely due to enhanced polarisation and the presence of additional hydrogen bonds. In contrast, this effect is reversed for malvidin 3,5-*O*-diglucoside, in which the addition of the C5 glucose sterically hinders self-association.

### 4.2. Effect of Co-Pigmentation on Colour

The effects of co-pigmentation can be divided into three categories: the first two involve the selective stabilisation of the coloured species. These manifest as an hyperchromic shift (Figure 4A) due to favourable interaction between the coloured anthocyanin species and the co-pigment, which depletes the available pool of free unbound anthocyanins and drives the equilibrium towards the formation of more coloured species. Co-pigmentation also enhances the resistance of the chromophore to hydration at C2 (which leads to the formation of colourless products) by steric hinderance, increasing colour half-life and reducing colour fading.

The third effect of co-pigmentation is the bathochromic shift (Figure 5B), caused by distortions in the electron orbitals between closely interacting aromatics [24]. Light is an electromagnetic wave, with an alternating dipole which is proportional to its wavelength. Relative to the strength of the electromagnetic field, atomic nuclei are too dense to experience any substantial polarisation, but electrons which are comparatively small experience a polarising effect when exposed to electromagnetic fields. As electrons carry a negative charge and oscillate in electromagnetic fields (harmonic oscillation, Figure 5), they induce dynamic polarisation in the molecular orbital. Electrons that oscillate at the same frequency as an incident photon (termed degenerate) can absorb energy from those photons, promoting them to a higher energy state (from HOMO–LUMO). This additional energy is radiated to the surroundings within picoseconds through a number of mechanisms as the electron moves back into the ground state [15].

The changes in electron density between the excited and ground state cause a change in the molecule’s dipole moment, and therefore the energy absorbed corresponds to the frequency of oscillation between these two different dipole moments, termed the transition dipole moment (*μ_ge_*, Figure 5B). The transition dipole is proportional to the polarisability of the compound (Figure 5C), and therefore distortion of the chromophore electron orbitals by the co-pigment decreases the energy gap, causing a bathochromic shift diagrammed in Figure 5C.

In particularly strongly associated pigment:copigment complexes, interactions result in the formation of a through-space charge transfer excitation state. This occurs when there is partial mixing of the π orbitals of the two interacting molecules. After photoexcitation to a higher energy state, the molecule then rearranges the electron density to a lower energy state. This reduces the excitation energy gap between the highest occupied molecular orbital (HOMO) and the lowest unoccupied molecular orbital (LUMO), and increases the absorption wavelength maxima, shifting the colour towards blue [23]. The local excitation state is still present, so this charge transfer band appears as a shoulder peak in the UV-VIS spectra.

### 4.3. Types of Co-Pigment

Owing to their large and rigid, polarisable surface, anthocyanins can theoretically form vertical stacking assemblies with over ten-thousand compounds [25]. However, the major natural co-pigment compounds identified are the hydrolysable tannins, flavonoids (such as flavones, flavonols, and other anthocyanins), and phenolic acids (Figure 6).

The basic requirement for a co-pigment is the presence of a planar arene moiety, which favours interaction with the anthocyanin chromophore. Increasing the size of the aromatic system or the substituent groups can have profound effects on the degree of co-pigmentation [27]. Planar flavonols such as kaempferol, quercetin, and myricetin are conjugated over their A-, B-, and C-rings and thus have the highest affinity for co-pigmentation.

Hydroxycinnamic acid (HCA) derivatives such as *p*-coumararic, caffeic, ferulic, and sinapic acids are commonly observed as both inter- and intramolecular co-pigments, producing a similar degree of colour modulation to planar flavonoids [15]. Increasing substitution of the arene group with hydroxyl and methoxy substituents increases the thermodynamic binding constant, and the affinity for co-pigmentation increases along the series: *p*-coumarate < caffeic acid < ferulic acid < sinapic acid [28,29]. Despite the advances in understanding co-pigmentation interactions, it is still unclear whether this increased affinity for highly substituted groups is attributable to distortion in pigment electron density, or the formation of additional direct electrostatic interactions between the substituent groups and the anthocyanin chromophore, although it is highly likely that both mechanisms play a role.

### 4.4. Blues from Intermolecular Co-Pigmentation

Blue colouration by intermolecular co-pigmentation alone is not common, and when it occurs, it is usually the result of co-pigmentation of anthocyanins with glycosylated flavonols and flavones. The blue-purple-coloured flowers of cranesbill, *Geranium* “Johnson’s Blue”, *Geranium pratense*, and *Geranium sanguineum* all contain the same anthocyanins: malvidin 3-*O*-glucoside-5-*O*-(6-*O*-acetylglucoside) and a minor fraction of malvidin 3-*O*-(6-*O*-acetyl glucoside) [30]. The variation in colour between these varieties (Figure 7) results from differences in the molar concentrations of flavonol co-pigments kaempferol and myricetin 3-*O*-glucoside and 3-*O*-sophoroside, with the blue-coloured petals of *Geranium* ‘Johnson’s Blue’ and *G. pratense* containing significantly higher molar ratios of flavonols than *G. sanguineum* [30,31] (Figure 7).

Pigment-co-pigment complexes are weak interactions and are attenuated by heating, dilution, and changes in pH. The formation of these complexes requires high molar ratios of co-pigment relative to the concentration of anthocyanins, the ratio depending on the affinity of the co-pigment for the anthocyanin. Observations of co-pigmentation with flavonols and flavones demonstrate the highest stabilising effect on anthocyanin/co-pigment complexes at ratios of 1:2–1:20, whereas hydroxycinnamic acids require concentrations between 1:10 and 1:100 for comparable hyperchromic and bathochromic shifts [22,32].

Using genetic engineering, Noda et al. [33] were able to generate blue through the expression of the gene encoding UDP—glucose/anthocyanin 3’,5’-*O*-glucosyltransferase from butterfly pea (*Clitoria ternatea*) and the flavonoid 3’,5’-hydroxylase from Canterbury bells (*Campanula medium*) in chrysanthemum, which does not naturally have the capacity to make blue flowers (Figure 8), in part because it lacks a gene encoding flavonoid 3’,5’-hydroxylase and consequently the ability to make delphinidin-based anthocyanins. However, expression of flavonoid 3’,5’-hydroxylase alone in chrysanthemum produced only a purple/violet colour [34,35].

Expression of the gene encoding UDP—glucose/anthocyanin 3’,5’-*O*-glucosyl transferase resulted in the modification of delphinidin 3-*O*-(6”-*O*-malonyl) glycoside in order to produce delphinidin 3-*O*-(6”-*O*-malonyl) glycoside-3’5’-di-*O*-glucoside (ternatin C5). In isolation, the glycosylation of 3’ and 5’ hydroxyl groups on the B-ring of the anthocyanin caused a reddening of the anthocyanin with the λ_max_ in the visible range dropping from 527 nm for delphinidin 3-*O*-(6”-*O*-malonyl) glycoside to 511 nm for delphinidin 3-*O*-(6”-*O*-malonyl) glycoside-3’5’-di-*O*-glucoside at pH 2.7. In fact the bluing of the chrysanthemum flowers was due to an increased capacity for co-pigmentation with the flavones luteolin 7-*O*-(6”-*O*-malonyl) glucoside and tricetin 7-*O*-(6”-*O*-malonyl)glucoside, which are present naturally in chrysanthemum flowers, which turned from violet in the presence of delphinidin 3-*O*-(6”-*O*-malonyl) glycoside to blue in the presence of delphinidin 3-*O*-(6”-*O*-malonyl) glycoside-3’5’-di-*O*-glucoside. The bathochromic shift of λ_max_ in the visible range was greater for the shift from cyanidin to delphinidin-based anthocyanins than for the di-glycosylation of the B-ring. However, delphinidin 3-*O*-(6”-*O*-malonyl)glycoside-3’5’-di-*O*-glucoside in combination with the flavonol co-pigments showed a strong shoulder of absorption at 590 nm, giving the bluing effect of co-pigmentation (Figure 8). For the delphinidin-based anthocyanins, co-pigmentation with flavonols increased the relative abundance of quinonoid base in the multistate equilibrium, whereas the abundance of anionic quinonoid base increased in the equilibrium of delphinidin 3-*O*-(6”-*O*-malonyl)glycoside-3’5’-di-*O*-glucoside with its flavonol co-pigments [33].

In nature, intermolecular co-pigmentation is dependent upon the co-localisation of the co-pigment with the anthocyanins in the vacuole of petal cells. For this reason, flavonol and flavone co-pigments in flowers are always glycosylated since glycosylation provides the signal for their transport to the vacuole. For other co-pigments such as caffeoyl and p-coumaroyl quinic acids in blue hydrangeas, accumulation in vacuoles does not involve glycosylation signals [26].

Most blue flowers produce anthocyanins that are stabilised by additional co-pigmentation interactions, which fall into two categories, namely, intramolecular association and metal cation chelation.

### 4.5. Intramolecular Association

Anthocyanins show a wide array of chemical diversity; of the nearly 700 species reported, close to 300 are acylated with one or more aromatic or aliphatic acyl groups, while approximately 100 of these contain both [36]. Although the acylated anthocyanins which produce the blue colours of flower petals are structurally diverse, the predominant anthocyanins are usually delphinidin derivatives.

Anthocyanins which are acylated with aromatic hydroxycinnamic (HCA) derivatives form non-covalent intramolecular associations with their covalently linked HCA moieties, which fold over the chromophore, protecting the chromophore from hydration [37] (Figure 9).

These interactions can be loosely categorised into two types: type 1 where the anthocyanin forms intramolecular co-pigmentation interactions with HCAs both above and below the chromophore, and type 2 where the anthocyanin forms intramolecular interactions on only one plane, leaving one plane exposed for the formation of self-association interactions with other anthocyanins, or co-pigmentation with a non-anthocyanin co-pigment [25] (Figure 10).

As these co-pigments are covalently linked (intramolecular) their effects on colour and stability are not dependent on concentration. As an example, the polyacylated anthocyanin pelargonidin 3-*O*-(2”(caffeoylglucosyl)-6”-(caffeoylglucosylcaffeoyl) glucoside)-5-*O*-glucoside produces a bathochromic shift compared to the non-acylated anthocyanin equivalent to the addition of external caffeic acid co-pigment at a concentration of 0.6M [38].

These types of interactions are dependent on specific structural features of the anthocyanin which, as a consequence of the diversity of aromatically acylated anthocyanins and potential co-pigments, are incompletely defined. In general, for HCA moieties linked to either the B- or C-rings of the anthocyanidin by monosaccharides or disaccharides, intramolecular co-pigmentation (as observed in ternatins, heavenly blue anthocyanin (HBA) from Japanese morning glory, and gentiodelphin) is more common than self-association. Anthocyanins with extended acyl residue linkers or repeating sugar–HCA sequences linked to sugars attached to the A- or C-rings favour self-association (as observed with tecophilin) [25].

The position of linkage of glycosyl–HCAs to the anthocyanin core imposes some geometric constraints, influencing how well the co-pigment can align with the anthocyanin chromophore, and hence the strength of the association (as measured by the degree of bathochromatic shift). The investigation of mono-acylated anthocyanins has led to the proposal of the following ranking: 3’~5’ > 7 > 3 [25].

Similarly, the position of acylation on the sugar can influence stacking interactions. Acyl groups are usually bound to the C6”-OH of the primary sugar, allowing enough bond flexibility for efficient stacking [39]. Acyl groups have been observed less frequently at C2”-OH, C3”-OH and C4”-OH groups of the primary sugar than on the C6” OH group [40]. In contrast to acyl groups linked to primary sugar OH groups, acylation of the secondary sugar on groups such as at C4”-OH appear to not interact with the chromophore at all [41].

Studies of di-acylated anthocyanins from red cabbage have shown that addition of a single sinapoyl group to the C6” position of first sugar (G1) on C3 of cyanidin 3-*O*-sophorose-5-*O*-glucoside causes a bathochromic shift of 10 nm in λ_max_ in the visible range, while addition of a second sinapoyl group on C2” of G2 gives a bathochromic shift of 24 nm in λ_max_. The equivalent bathochromic shifts that could be achieved with addition of free sinapic acid to cyanidin 3-*O*-sophorose-5-*O*-glucoside and cyanidin 3-*O*-(6”-sinapoyl) sophorose-5-*O*-glucoside were 4 and 11 nm, respectively [39], demonstrating the effectiveness of intramolecular co-pigmentation compared to intermolecular co-pigmentation.

Acylated anthocyanin complexes have increased colour stability and generally increased aromatic acylation increases blueness and anthocyanin stability. The addition of water to C2 of the flavylium ion is key to fading of anthocyanins because it causes the formation of colourless hemiketal epimers. By promoting the formation of quinonoid bases in the multistate equilibrium, HCA decoration increases anthocyanin stability because quinonoid bases cannot undergo hydration reactions. Aromatic acylation also increases the strength of π stacking interactions and slows irreversible colour loss by chemical degradation involving cleavage of the C-ring, although these latter processes are not well understood [39]. Consequently, aromatic acylation causes big increases in the stability of anthocyanin pigments, as illustrated in Figure 11 for extracts of di-acylated anthocyanins from the desert bluebell (*Phacelia campanularia*) and quadri-acylated anthocyanins from butterfly pea (*Clitoria ternatea*) compared to the non-acylated cyanidin 3-*O*-rutinoside.

### 4.6. Aggregation of Aromatically Acylated Anthocyanins

Depending on the position of HCA attachment and the presence of additional stabilising glycosylation, some acylated anthocyanins aggregate, even at low concentration, at pH values greater than 2 and less than 8. Studies of aggregation undertaken at elevated temperatures have suggested that aggregation is the result of intermolecular rather than intramolecular associations [42]. Aggregation of this type can occur naturally in plant cells giving rise to the formation of anthocyanic vacuolar inclusions (AVIs) [43,44]. The production of aggregated acylated anthocyanins in AVIs is used in some flowers to create dark spots of colour, as seen in the central region of the sepals of lisianthus (*Eustoma grandiflora*). In the case of lisianthus, it is the absence of additional glycosylation of the C3 of delphinidin and cyanidin 3-*O*-galactoside-5-*O*-(*p*-coumaroyl) glucoside and 3-*O*-galactoside-5-*O*-(feruloyl)glucoside on the inner epidermis of the base of the sepals that is associated with AVI formation compared to the delphinidin and cyanidin-3-*O*-(rhamnosyl) galactoside-5-*O*-(*p*-coumaroyl) glucoside and 3-*O*-(rhamnosyl) galactoside-5-*O*-(feruloyl) glucoside.

These observations have been interpreted in terms of reduced glycosylation favouring AVI formation, which fits well with in vitro observations that 5-*O*-glycosylation reduces AVI formation in vivo [44]. The role of AVI formation in patterning the flowers of lisianthus is illustrated in Figure 12.

Anthocyanins from blue flowers commonly also contain aliphatic acyl groups, such as malonic acid. Addition of malonyl groups does not impact the colour of anthocyanins directly [45]. The contribution of aliphatic acyl groups to stacking interactions (and hence indirectly to colouration) is unclear, although a few functions have been proposed. Acylation with a malonyl group was observed to lower the pK_a_ of anthocyanins in *Matthiola incana*, possibly by stabilising the neutral species through the formation of hydrogen bonds between the malonyl group and the oxygen at C7 of the quinonoid base [25]. Acylation with multiple aromatic groups reduces the solubility of anthocyanins in the aqueous cell sap. This may pose a problem during the biosynthesis of poly-acylated anthocyanins, such as the ternatins—the largest of which is acylated with four *p*-coumaroyl moieties. Therefore, acylation with aliphatic organic acids on glucose moieties at C3, C5, or C7 may help keep these anthocyanins soluble in the aqueous cell sap during biosynthesis [44,45].

### 4.7. Fuzzy Metal Complexes

Anthocyanins such as cyanidin, delphinidin, and petunidin, with two free hydroxyl groups on the B-ring, demonstrate a colour change upon chelating multivalent metal cations. 

The metal cation competes with the hydrogen ions and the red flavylium cations are transformed to form the blue quinonoid base anions [46]. The complexes are then stabilised through typical co-pigmentation interactions with external co-pigments, or through self-association with other anthocyanins, where both the co-pigment and anthocyanin are coordinated by the metal cation.

These complexes still require high relative molar concentrations of co-pigment, and are blue only in aqueous solution, where their colour is similarly attenuated by dilution. Further, their colour is lost upon attempts to crystallise or isolate these pigments. For this reason, they are referred to as “fuzzy metal complexes”, due to the difficulties in clarifying the mechanisms of association [31].

The best studied cases of fuzzy metal complexes are those present in *Hydrangea macrophylla*. These flowers range in colour from red to mauve to blue, and all contain delphinidin 3-O-glucoside, co-pigments (mainly 3-O-caffeoyl and 3-O-p-coumaroyl quinic acids), and Al^3+^ in differing concentrations (Figure 13).

Neither Al^3+^ or 3-O-caffeoylquinic acid alone could produce blue colour when mixed with delphinidin 3-O-glucoside, the anthocyanin present in both blue and red hydrangeas, in the molar ratios found in the flower sepals. The blue variety, however, had significantly higher concentrations of co-pigment and Al^3+^ compared to the red variety [48]. In an elegant recent publication, it was shown that the anthocyanin accumulates in the sub-epidermal layer of sepals of both red and blue varieties. The concentration of caffeoyl/*p*-coumaroyl 3-*O*-quinic acids, which serve as co-pigments, are high in several cell layers of the sepals of hydrangea. However, it is the high levels of aluminium in the subepidermal layer that confer the specificity of blue, achieved through the association of anthocyanins, co-pigments, and Al^3+^ ions in the subepidermal layer of blue flowers (Figure 13). In the comparator red-flowered variety, levels of Al^3+^ were found to be much lower in the subepidermal cell layer of the sepals [26].

Historically, this mechanism has been exploited to glean structural information about the anthocyanin under investigation, the addition of Al^3+^ producing a characteristic blueing of the anthocyanin indicating the presence of two free hydroxyl groups present on the B-ring. Gardeners have used hydrangea flower colour to demonstrate the effect of soil pH on flowers; plants grown on acid soils are bluer because low pH results in more bioavailable Al^3+^ to form the fuzzy metal complex. Cultivation of the same varieties on more alkaline soils results in red flowers due to the low availability of Al^3+^ under these conditions [49,50]

### 4.8. Metalloanthocyanins

A specific subset of metal-coordinated anthocyanins form large supramolecular structures termed metalloanthocyanins, which have been demonstrated to perform an important role in the development of blue colour in several floral systems. These complexes are composed of stoichiometric amounts of anthocyanins, flavones, and metal ions. For all metalloanthocyanins, these ratios are fixed at 6:6:2, respectively [31].

The formation of these complexes requires similar structural properties to those seen in the more typical metal-coordination with anthocyanins—two unsubstituted neighbouring hydroxyl groups on the B-ring. Both the anthocyanins and flavones self-associate in pairs with distances between the aromatic rings of the respective pair of approximately 3.3 Å, indicative of hydrophobic interaction [51] (Figure 14).

To date, only a few metalloanthocyanins have been identified with colours ranging from purple to blue. These known metalloanthocyanins consist of either delphinidin, cyanidin, or petunidin and Mg^2+^, Al^3+^, or Fe^3+^ as metal cations centralised within the complex [31]. Generally in nature, cyanidin-based anthocyanins tend to form complexes with Fe^3+^, whereas delphinidin-based anthocyanins form complexes with Mg^2+^, although there are exceptions to this “rule”. Of course, in vitro these predominant metals may be capable of being substituted by others (unavailable in nature), which may impact the colour and stability of the complex [52].

## 5. Forces Involved in Co-Pigmentation

Hunter and Sanders [53] originally proposed that the π electron density of an unsubstituted aromatic ring results in a quadrupole moment, with a partial negative charge above both aromatic faces, and partial positive charge around the periphery. π-interaction is an attractive force between electrical quadrupoles, overpowering the repulsion of π-electron clouds. Where aromatic systems share a similar electron density, σ–π interactions are favourable and should therefore favour an edge-to-face or off-centre parallel arrangement, disfavouring face-centred arrangements (Figure 15).

The addition of electron-withdrawing substituents to an aromatic (arene) polarises π electron density away from the aromatic centre, reversing the direction of the quadrupole [23]. Pairs of electron-rich and electron-deficient aromatics preferentially pair in a face-centred arrangement, referred to as “aromatic donor–acceptor interaction” (Figure 15).

Pigment–co-pigment interactions are often attributed overwhelmingly to π–π interactions. However, these fail to account for other molecular forces which can initiate and drive co-pigmentation. The magnitude of the interaction is the product of all forces acting between the two aromatic systems, maximising electrostatic forces and dispersive attraction.

π–π interactions are not limited to arene moieties; cyclical aliphatic molecules have also been demonstrated to take part in these interactions, suggesting that the requirement for electron delocalisation (as seen in aromatics) is not strict [54]. This is particularly noteworthy within non-aromatic heterocycles, such as glucose, where the presence of an oxygen atom introduces asymmetry within the heterocycle, resulting in polarisation. Within any anthocyanin or co-pigment, there will be regions of high- and low-electron density. Therefore, the overall strength of intra- and intermolecular co-pigmentation interactions will depend on the alignment of these regions [53].

Such interactions may also explain why the addition a single glucose moiety at C3’ and C5’ (to produce ternatin C5) was sufficient to generate blue colouration in butterfly pea and also in genetically modified chrysanthemum [29,33]. These glucose groups may increase the dispersive attraction, and possibly direct electrostatic interactions, pulling the co-pigment closer to the chromophore and reducing the transition energy gap, thus producing a bluer colour (Figure 8).

### Hydrophobic Interactions

Anthocyanins form the strongest co-pigmentation interactions when dissolved in polar solvents and there is a decrease in magnitude of the co-pigmentation effect in non-aqueous solvents, where co-pigmentation has been attributed largely to hydrophobic interactions [20].

These hydrophobic interactions are both classical (entropic) and nonclassical (enthalpic). Water molecules solvating anthocyanins and their co-pigments form a highly organised hydrogen bond network around the aromatic intramolecular region (hydrophobic cavity) [25]. As the pigment and co-pigment come together, the hydrogen bonds between water molecules in the solvation shell are broken and released from the hydrophobic cavity, increasing the system entropy and enthalpy (Figure 16). The barrier to complex formation is further reinforced by the required decrease in the system entropy (requiring energy) as the acyl group rotates about the C6” axis to form a highly ordered structure.

However, all observed co-pigmentation interactions are spontaneous at room temperature. To account for the energy deficit from the initial bond-breaking, energy is counterbalanced by a decrease in enthalpy by the formation of new hydrogen bonds between the liberated water molecules. In all cases investigated, pigment-co-pigment complexes have been exothermic [55].

Hydrophobic effects are not strictly necessary for interactions to occur. There have been many observations of anthocyanins forming pigment-co-pigment interactions in non-polar solvents, where hydrophobic forces would be minimal [56]. Hydrophobic forces may drive the arene moieties close enough for dispersive and electrostatic attraction, which then dictate the final spatial arrangement and hence, the degree of charge transfer resulting in observed colour changes.

## 6. How Can We Improve Natural Blue Food Colourants?

One of the greatest barriers to the deployment of natural blue colourants for food is the relative cost of anthocyanins and the glycosylated flavonoids necessary for formulations of intermolecular co-pigments compared to the cost of synthetic blues. The cost of purified anthocyanins themselves, as used in medical practice and research, means that these purified products are non-competitive with synthetic food colourants. In addition, regulations governing food colourants state that any pigments extracted from foods must be listed by E-number or have received listed approval by the Food and Drug Administration (FDA) in the USA, rather than being referred to by their more familiar and consumer-friendly names. This has led to many natural food colourants being developed using minimal extraction procedures, as “colouring foods”, although FDA regulations exclude such products for use only or principally as colourants. Even when used as colouring foods, the costs are relatively high compared to the costs of synthetic colourants, especially blues. In addition, anthocyanin-based natural blues suffer from instability, limiting use in processed foods.

Because the term “natural” is paramount in the natural colourants industries, to distinguish these products from synthetic colourants (which have declined in popularity and, in several cases, have been outlawed on the grounds of health concerns), “biotechnological” approaches to improving availability of natural blue colourants are largely focused on scale-up from specific natural sources and improvements in formulations. Scale-up has been possible from black or purple carrots, purple sweet potato, and blueberries, because such plants are already cultivated widely and, for some, colour extraction may offer sustainable ways of using waste material from mainstream commercial use, such as the use of skins of red grapes from the wine industry for colourant production. However, while providing good sources of red, brown, and even purple colourants, these approaches are not feasible for strong blues. In confectionary, the multiply aromatically acylated anthocyanins of red cabbage have been of considerable interest because they offer a scalable source of pigments with reasonable stability between neutral and high pH. Extraction of pigments from *Brassicaceous* species always carries the risk of accompanying undesirable tastes, however, as was discovered when “Smarties” first switched to natural red colours extracted from radishes.

Several large colourant industries have toyed with ideas of creating “natural” or nature-equivalent blues by reconstituting metal complexing with anthocyanins. However, if iron is used (which could have accompanying consumer benefits in addressing dietary iron deficiencies), there can be accompanying problems with undesirable tastes, in addition to the cost of iron fortification, as well as the high costs of associated co-pigments required for effective formulation of blues.

Consequently, the most effective approach to identifying better natural blue food colourants may be to continue screening blue flowers and fruits to find the anthocyanins with the best tones over a wide range of pH values and good stability, of the order of months rather than days, under conditions used for cooking and processing. Such candidates include the ternatin extracts produced from the flowers of butterfly pea, which have remarkable stability even at high pH (Figure 17C) and in cooked goods compared to spirulina natural blue and the red betacyanins of beetroot natural food colourants (Figure 17F).

The major issue with using butterfly pea as a source of natural blue is scale-up. Plants are not hugely productive in terms of flower numbers and the flowers are ephemeral, lasting little longer than a day with their full blue colour. However, dried butterfly pea flowers are used traditionally as a herbal tea in Southeast Asia, and thus have a history of safe consumption, with accompanying anecdotal health benefits. In addition, as a legume, butterfly peas could be beneficial in fertilising marginal soils where they often grow wild.

Other biotechnological approaches to improved production of natural blue colourants could involve genetic modification of plant cell cultures so that they constitutively produce high levels of anthocyanins. Cell cultures have been used for colour production, but invariably suffer from long-term instability. Stable, high-level anthocyanin production has been demonstrated to be feasible in tobacco cells which have been genetically engineered to produce acylated anthocyans as well as non-acylated forms in large amounts using transcription factors that induce anthocyanin biosynthesis [57]. Similar strategies could be used to produce high levels of co-pigments, both flavonols and flavones, for in vitro formulations of natural blues with improved performance as colourants.

Probably the best chassis for the highly decorated anthocyanins necessary for producing strong and stable natural blues are species such as butterfly pea itself, whose genome encodes all the enzymes necessary to make quadri-acylated delphinidin-based anthocyanins. Suspension cultures would need only to be transformed with the genes encoding well-characterised regulators of anthocyanin biosynthesis to develop effective scale-up production systems. However, such strategies involve genetic modification, and although production systems for high value pharmaceuticals such as paclitaxel make use of non-GM cell cultures, it is perhaps unlikely that the natural colourant industry would embrace readily genetically modified cell suspension culture production systems, even though they may offer the most sustainable and cost-effective production systems for natural blues.

## Figures and Tables

**Figure 1 plants-10-00726-f001:**
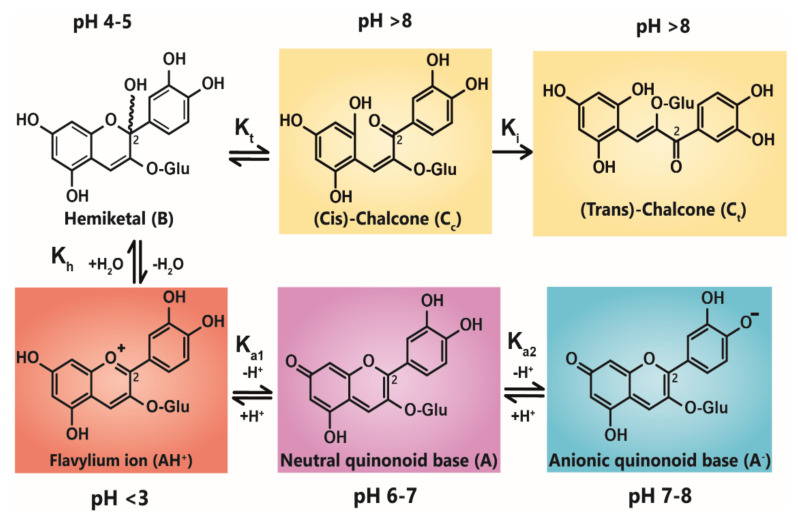
Structural transformation of cyanidin-3-*O*-glycoside as a function of pH. The three cyanidin species are given with their expected colours as outlined. Deprotonation from the C5-OH has been omitted for simplicity.

**Figure 2 plants-10-00726-f002:**
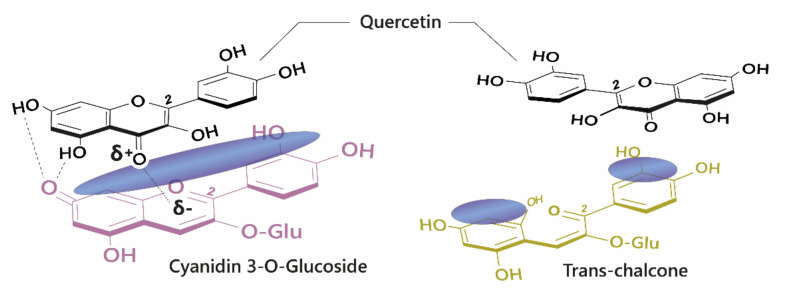
**Left:** Co-pigmentation interaction between cyanidin 3-*O*-glucoside and quercetin. Blue clouds above the structures denote polarisable surface area. Direct electrostatic interactions form between substituent groups, stabilising the coloured species. **Right:** Ring opening breaks the conjugated system across the tricyclic ring, reducing the planar surface for co-pigmentation.

**Figure 3 plants-10-00726-f003:**
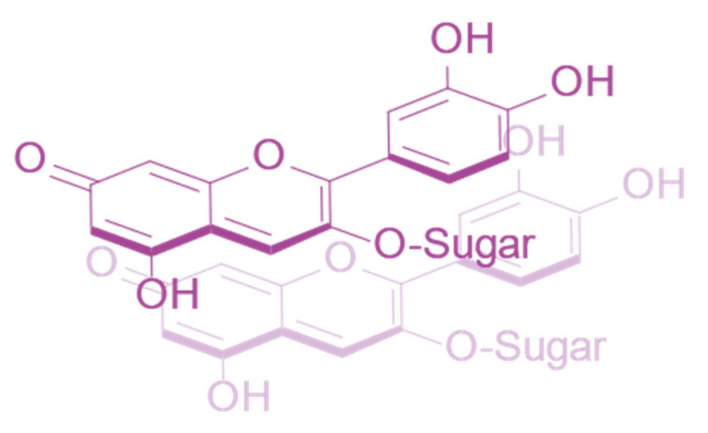
Self-association interaction between two cyanidin-3-*O*-glucoside quinonoid bases.

**Figure 4 plants-10-00726-f004:**
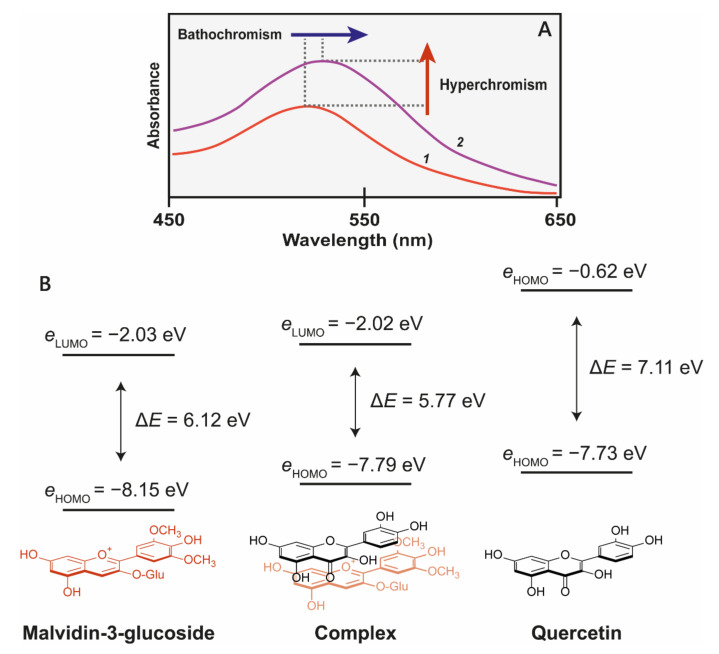
(**A**) Co-pigmentation of malvidin 3-glucoside with quercitin at pH 3.6. 1 = 500 μM malvidin 3-glucoside, 2 = 500 μM malvidin 3-glucoside + 1 mM quercitin [22]. (**B**) The molecular orbital correlation diagram of malvidin 3-*O*-glucoside and quercetin shows the energy changes as a result of co-pigmentation complex formation. Reduction of highest occupied molecular orbital of chromophores (HOMO)–lowest unoccupied molecular orbital (LUMO) energy gap (ΔE) results in a bathochromatic shift [23].

**Figure 5 plants-10-00726-f005:**
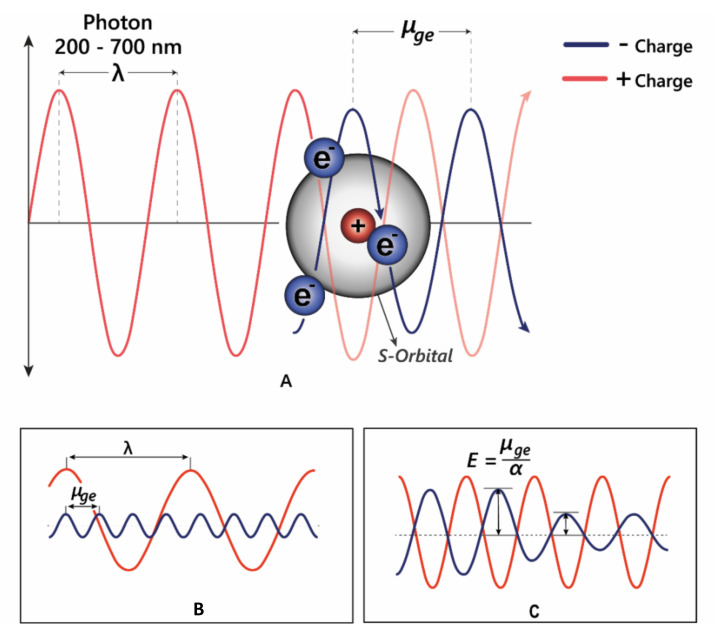
(**A**) Electrons can oscillate between the HOMO–LUMO by absorbing specific wavelengths of light. (**B**) The transition dipole moment (*μ_ge_*) is the difference between the energy in the ground state and the excited state. For promotion of an electron to a higher molecular orbital, *μ_ge_* = λ. (**C**) The strength of the interaction between electrons and electromagnetic radiation is dependent on the polarisation—α; in highly polarised systems such as the anthocyanins, the transition energy is reduced, allowing absorption of longer wavelength light.

**Figure 6 plants-10-00726-f006:**
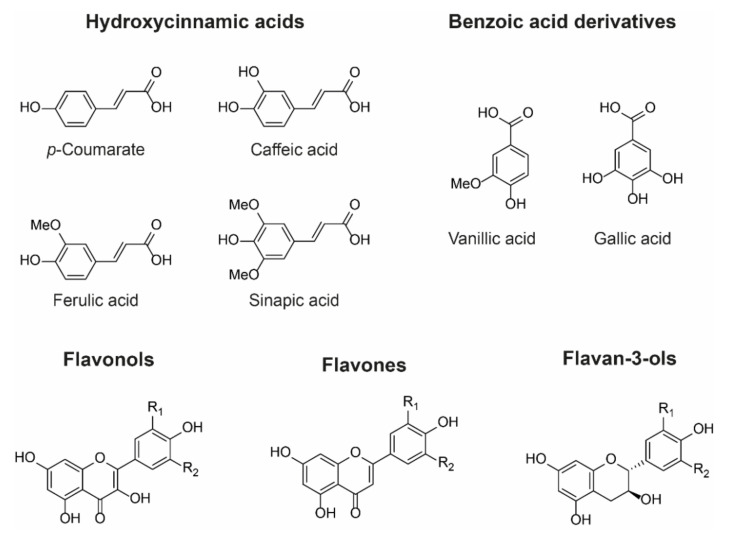
Examples of chemical structures of commonly observed co-pigments. In nature, co-pigmentation is dependent on co-localisation of anthocyanins and co-pigments within the vacuole. Consequently, flavone and flavonol co-pigments are glycosylated in order to facilitate their transport to the vacuole. Hydroxycinnamoyl quinic acids such as 3-*O*-caffeoyl and 3-*O*-*p*-coumaroyl quinic acids can be transported to the vacuole as aglycones and are found as co-pigments in hydrangea flowers, for example [26].

**Figure 7 plants-10-00726-f007:**
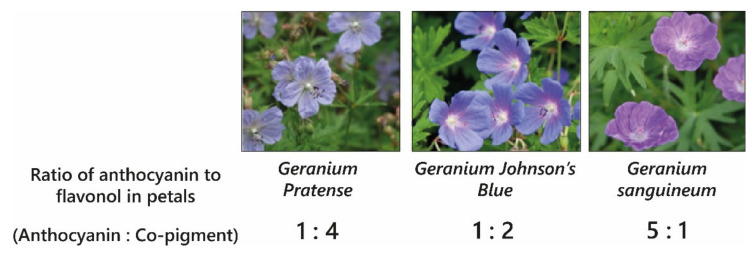
The different colours of varieties of cranesbill are determined by intermolecular co-pigmentation between malvidin 3-*O*-glucoside-5-*O*-(6-*O*-acetylglucoside) and varying relative amounts of flavonol co-pigments kaempferol and myricetin 3-*O-*glucoside and 3-*O*-sophoroside [30].

**Figure 8 plants-10-00726-f008:**
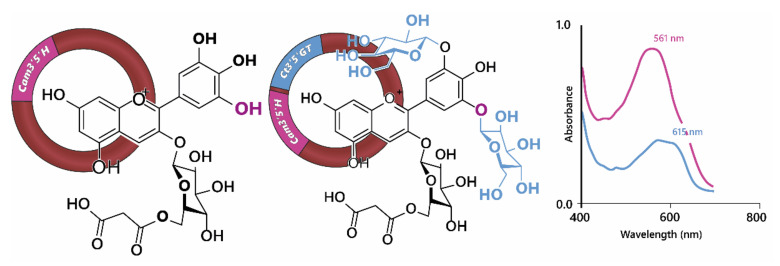
Noda et al., [33] achieved blue colours in chrysanthemum through the expression of *Cm3’5’H* (flavonoid 3’,5’-hydroxylase) and *Ct3’5’GT* (UDP—glucose/anthocyanin 3’,5’-*O*-glucosyltransferase). Hydroxylation of cyanidin 3-*O*-(6”-*O*- malonyl) glucoside at 5’ was not sufficient to generate blue colours, requiring the addition of two glucose groups to the B-ring, demonstrated by the shift in visible λ_max_ shown on the right.

**Figure 9 plants-10-00726-f009:**
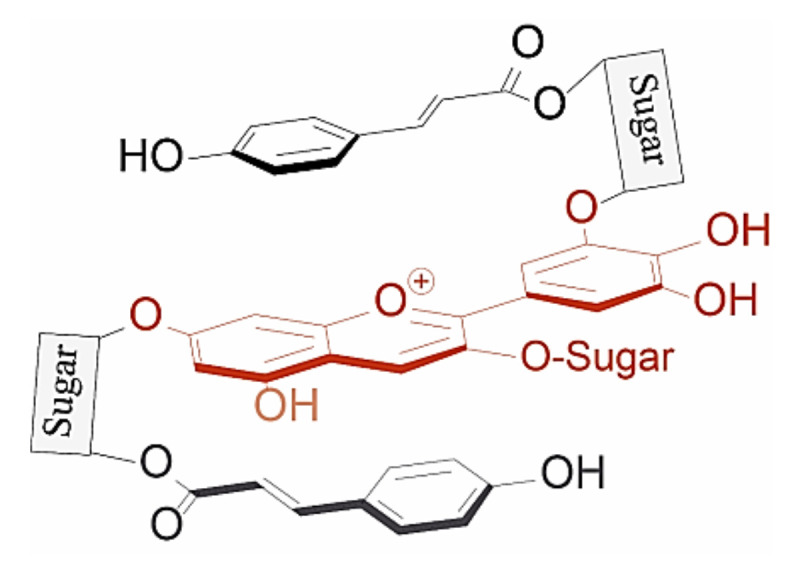
Typical structure of poly-acylated anthocyanins allowing non-covalent interaction between the anthocyanin chromophore and covalently linked acyl moieties (intramolecular co-pigmentation) [25].

**Figure 10 plants-10-00726-f010:**
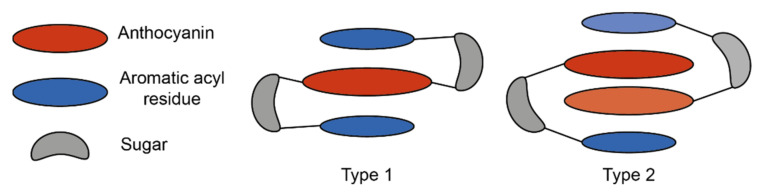
Simplified schematic of the two major types of intramolecular co-pigmentation. Type 1 complexes form intramolecular co-pigmentation interactions above and below the anthocyanin chromophore. Type 2 complexes form only one intramolecular interaction, and self-association interactions on the reverse plane. Adapted from [31].

**Figure 11 plants-10-00726-f011:**
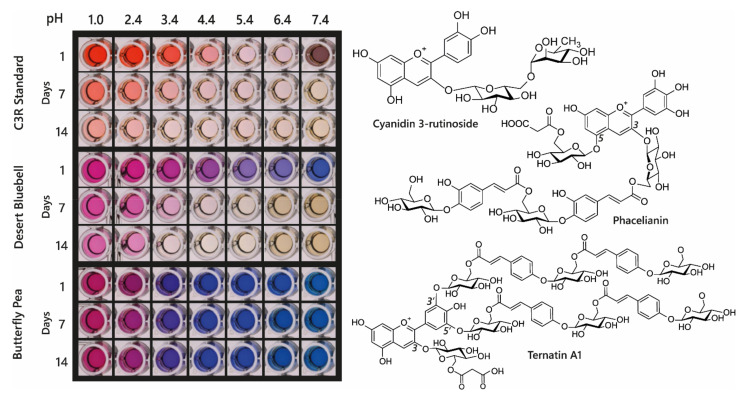
Demonstration of increasing blueness and increasing stability associated with multiple acylation of anthocyanins. Cyanidin 3-*O*-rutinoside was purchased commercially, and anthocyanins were extracted from desert bluebell (*Phacelia campanularia*) and butterfly pea (*Clitoria ternatea*). The most highly acylated anthocyanins in each extract are illustrated to the right. The plates show the colours of the anthocyanin extracts at different pH values and following storage at room temperature for 1, 7, and 14 days.

**Figure 12 plants-10-00726-f012:**
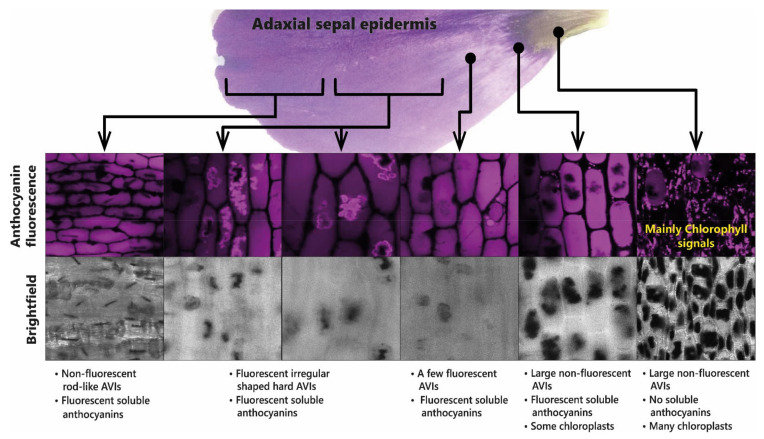
Presence of anthocyanin aggregates in the sepals of lisianthus is associated with the production of aggregates of acylated anthocyanins. The anthocyanic vacuolar inclusions (AVIs) that give the dark colour to the base of the sepals do not fluoresce under UV light. The soluble anthocyanins in lisianthus sepals fluoresce as do some additional anthocyanin aggregates from the outermost regions of the sepals.

**Figure 13 plants-10-00726-f013:**
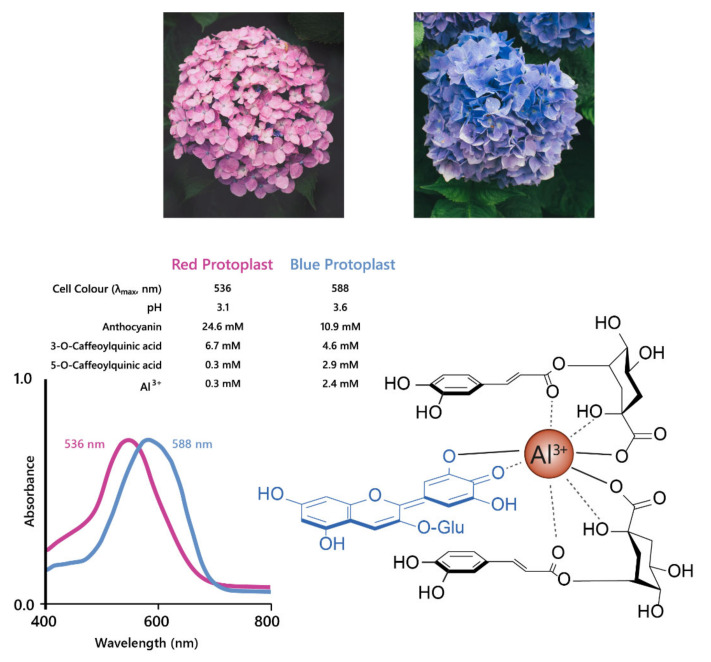
Hydrangeas produce blue flowers due to the formation of metal chelates between delphinidin, co-pigments, and Al^3+^. This complex stabilises the anionic quinonoid base species, producing a bathochromic shift of 52 nm, as shown with the shift in λ_max_ for the visible absorption spectra shown on the left. Adapted from [47].

**Figure 14 plants-10-00726-f014:**
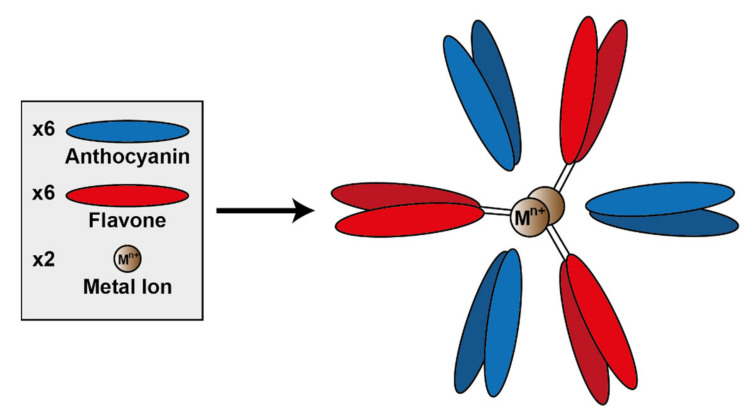
Assembly of metalloanthocyanin complex by self-assembly of six anthocyanin molecules, six flavone molecules, and two metal cations. Adapted from [31].

**Figure 15 plants-10-00726-f015:**
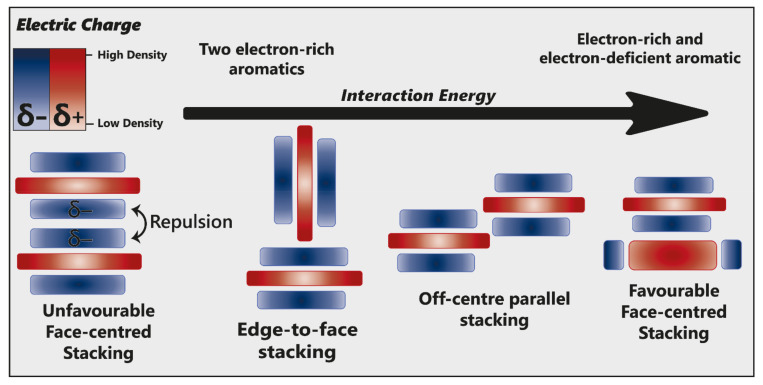
Alternative modes of stacking are shown, emphasising the locations of electrostatic attraction or repulsion. As the interaction energy increases, so too does complex stability. Adapted from [23].

**Figure 16 plants-10-00726-f016:**
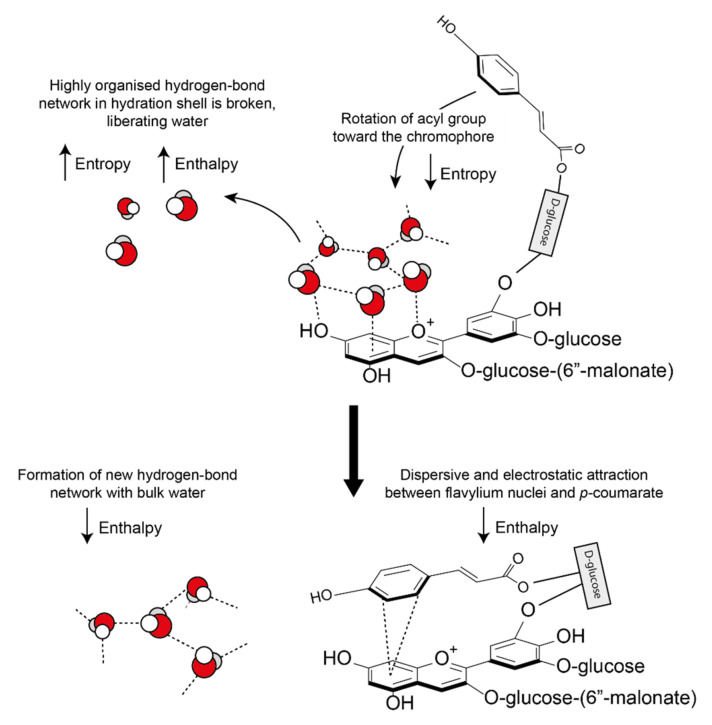
Classical and non-classical hydrophobic forces involved in the formation of co-pigmentation complexes. As the two interacting aromatics approach each other, hydrogen bonds between the highly organised hydration shell are broken, increasing both enthalpy and entropy. New hydrogen bonds form between the displaced water molecules and the bulk water, decreasing enthalpy. A further decrease in enthalpy is observed due to the formation of attractive dispersive and electrostatic interactions. All observed co-pigmentation interactions are exothermic (−ΔH) and spontaneous (−ΔG0) at room temperature (23 °C).

**Figure 17 plants-10-00726-f017:**
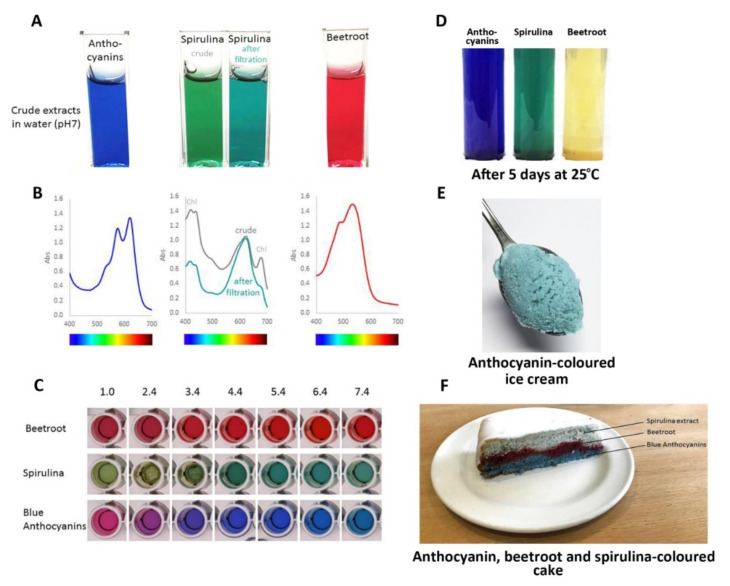
Comparison of performance of natural colours from different botanical sources: blue from flowers of butterfly pea, commercial natural “blue” from spirulina, and red from beetroot. (**A**) The colours of these colourants following fresh extraction/preparation in water at pH 7.0. The spirulina extract changed colour slightly following filtration to remove non-water-soluble material. (**B**) The visible absorption spectra of these colourants. Filtering of the spirulina extract removed some of the material absorbing at shorter wavelengths. (**C**) The effect of pH on the three colourants. The blue anthocyanins from butterfly pea did not maintain their “blueness” below pH 3.0, but provided strong blues over the range pH 3.0–8.0. (**D**) Stability of colours following the storage of fresh extracts at room temperature for 5 days. The red colour conferred by the betacyanins of beetroot showed poor stability. (**E**) Blue ice cream coloured by an extract from flowers of butterfly pea. (**F**) The effects of cooking on colour stability. The colour of the spirulina extract was particularly sensitive to high temperature.

**Table 1 plants-10-00726-t001:** General structures of common anthocyanins.

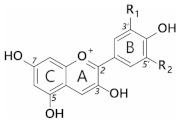		
Anthocyanidin	R1	R2
Pelargonidin	H	H
Cyanidin	H	OH
Delphinidin	OH	OH
Peonidin	OCH_3_	H
Petunidin	OH	OCH_3_
Malvidin	OCH_3_	OCH_3_

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
