# Peer review of "Natural Blues: Structure Meets Function in Anthocyanins"

_plants, 2021, doi:10.3390/plants10040726_

Round 1
Reviewer 1 Report
This review article by Houghton et al. addresses the complex physicochemical mechanisms conditioning the hue and stability of color of blue anthocyanins. The compiled data is of value for the food colorant industry and related fields of research, as chemically-stable natural colorants with blue hues that can be readily used as food dyes are scarce and difficult to achieve. The manuscript is very well written, and may be publishable in Plants after the following points are addressed. - Since there are several published reviews concerning the relationship between anthocyanins chemical structure and their function, I´d ask the authors to include an introductory paragraph clearly stating the need of this review and the differences with previews reviews. - In general, there is very little discussion on the genetics side of this topic. In Point 6 (How can we improve natural blue food colourants?), it would be desirable to have a more extensive discussion on what genes from the flavonoid pathway may be target candidates for using transgenic approaches for producing and/or increasing chemically-stable blue anthocyanins in plants.
Author Response
Reviewer 1:
This review article by Houghton et al. addresses the complex physicochemical mechanisms conditioning the hue and stability of color of blue anthocyanins. The compiled data is of value for the food colorant industry and related fields of research, as chemically-stable natural colorants with blue hues that can be readily used as food dyes are scarce and difficult to achieve. The manuscript is very well written, and may be publishable in Plants after the following points are addressed.
- - Since there are several published reviews concerning the relationship between anthocyanins chemical structure and their function, I´d ask the authors to include an introductory paragraph clearly stating the need of this review and the differences with previews reviews.
Response: We have added two sentences to the abstract highlighting our aims in the review which are to explain how structure determines the functionality of anthocyanin pigments, particularly their colour and their stability. Where possible we describe the impact of progressive decorations on colour and stability, drawn from extensive but diverse physicochemical studies. This emphasises the objectives of the review presented in the last paragraph of the introduction.
- - In general, there is very little discussion on the genetics side of this topic. In Point 6 (How can we improve natural blue food colourants?), it would be desirable to have a more extensive discussion on what genes from the flavonoid pathway may be target candidates for using transgenic approaches for producing and/or increasing chemically-stable blue anthocyanins in plants.
Response: Most of the genes responsible for creating blue anthocyanins are specific to individual plant species and impact colour in complex ways. We have given specific examples where appropriate, but thorough reviews of enzymes available for such modifications have been provided in excellent recent reviews by specialists in engineering flower colour, especially by Yoshinora Tanaka. Any transgenic approach to producing food colourants is fraught with difficulties, principally regulatory- and cost-associated and we have tried to highlight these issues jn the final section of the review. The truth is that there are already natural blue food colourants available, but unless the cost of production drops or the regulatory burdens on food colourants ease, biotechnology will not contribute to the production of natural blue colourants that can be used for foods. A low tech method of scale-up production of existing natural blues from flowers would appear a more feasible approach and we have summarised these issues in the final section of the review.
Reviewer 2 Report
This is a review paper summarizing the structural basis for anthocyanins as pigments. The topic fits well to the Special Issue.The paper is comprehensive and well-written and organized and is multidisciplinary from plant physiology (rather superficial but it's all right for this topic) to quantum chemistry. I have enjoyed reading it. The illustrations are clear (but see below) and informative. Therefore, without going into too much detail, I recommend the Editors of Plants to publish it pending some minor corrections that are, in my opinion, necessary to solidify the quality of this article.
So, first, I would recommend to modify the title, that while being brief and intriguing, does not really reflect the actual content. Perhaps replace "plant natural products" with anthocyanins, as simple as that. Or anything else that would let a potantial reader to know at first glimpse what to expect inside.
Second, in line 37-39, maybe it would be useful to mention also other pigment classes such as various quinones (by the way, much more widespread than the endemic betalains - which in turn, are important for food industry); some alkaloids also happen to be brightly colored; Also, (line45), correct the phrase - "betacyanins (a betalain found in Caryophyllales) to betacyanins (betalains found in some of the Caryophyllales)
3rd, lines 89-90, please, explain also that in living plants, anthocyanins like all flavonoids have their rings formed in fact another way - i.e by condensing a C6-C3 "unit" with 3 malonyl-CoA and so on. It is not exactly "formed" by fusion of a "benzopyrylium core with phenyl group attached to C2", is it?
4th. In figure 4, 7-sulfoquercetin is mentioned in legend while in the figure, we see unsubstituted quercetin. Please, correct it.
5th (really minor but should be) - a scale bar would be useful in all photographs showing a natural object, but at least in the microscopic images.
Author Response
Reviewer 2:
This is a review paper summarizing the structural basis for anthocyanins as pigments. The topic fits well to the Special Issue.The paper is comprehensive and well-written and organized and is multidisciplinary from plant physiology (rather superficial but it's all right for this topic) to quantum chemistry. I have enjoyed reading it. The illustrations are clear (but see below) and informative. Therefore, without going into too much detail, I recommend the Editors of Plants to publish it pending some minor corrections that are, in my opinion, necessary to solidify the quality of this article.
- So, first, I would recommend to modify the title, that while being brief and intriguing, does not really reflect the actual content. Perhaps replace "plant natural products" with anthocyanins, as simple as that. Or anything else that would let a potantial reader to know at first glimpse what to expect inside.
Response: This is an excellent suggestion, and we have changed ‘natural plant products’ to ‘anthocyanins’.
- Second, in line 37-39, maybe it would be useful to mention also other pigment classes such as various quinones (by the way, much more widespread than the endemic betalains - which in turn, are important for food industry); some alkaloids also happen to be brightly colored; Also, (line45),
Response: This is a review on plant natural products – anthocyanins - in a journal called Plants. We start the paragraph referred to by stating: ‘Plants produce an array of natural colourants, which can be broadly separated into four major groups:’. It seems misplaced to include quinones at this point because coloured quinones are derived almost exclusively from microbes, both bacteria and fungi. Although a few coloured alkaloids are produced by plants, they can not be classified as a ‘major’ group from plants. Therefore bearing in mind our topic, our selected readership and for clarity we have not included these other groups of compounds.
- correct the phrase - "betacyanins (a betalain found in Caryophyllales) to betacyanins (betalains found in some of the Caryophyllales)
Response: We have made this correction.
- 3rd, lines 89-90, please, explain also that in living plants, anthocyanins like all flavonoids have their rings formed in fact another way - i.e by condensing a C6-C3 "unit" with 3 malonyl-CoA and so on. It is not exactly "formed" by fusion of a "benzopyrylium core with phenyl group attached to C2", is it?
Response: We have rephrased the sentence to make clear the route of synthesis of anthocyanins by condensation of a C6-C3 compound (p-coumaroyl CoA) with 3 molecules of malonyl CoA.
- In figure 4, 7-sulfoquercetin is mentioned in legend while in the figure, we see unsubstituted quercetin. Please, correct it.
Response: An image for quercetin has been identified and used to replace the image for sulfoquercetin. Additional references citing the original publication have been added.
- 5th (really minor but should be) - a scale bar would be useful in all photographs showing a natural object, but at least in the microscopic images.
Response: Unfortunately, the images were not all taken at exactly the same magnification, and we have no record of what those magnifications were. The lack of a scale bar does not impact the information conveyed in the figure (each cell is about 20 μm across), and we feel that the figure conveys exactly the same information with or without a scale bar.
Reviewer 3 Report
The manuscript titled "Natural Blues: structure meets function in plant natural products" is carefully structured and well written. I want authors to congratulate authors for their effort.
Author Response
The manuscript titled "Natural Blues: structure meets function in plant natural products" is carefully structured and well written. I want to congratulate authors for their effort.
Response: Many thanks